# MeasureIce: accessible on-the-fly measurement of ice thickness in cryo-electron microscopy

Hamish G. Brown [1✉] & Eric Hanssen [1,2]

Ice thickness is arguably one of the most important factors limiting the resolution of protein structures determined by cryo-electron microscopy (cryo-EM). The amorphous atomic structure of the ice that stabilizes and protects biological samples in cryo-EM grids also imprints some additional noise in cryo-EM images. Ice that is too thick jeopardizes the success of particle picking and reconstruction of the biomolecule in the worst case and, at best, deteriorates eventual map resolution. Minimizing the thickness of the ice layer and thus the magnitude of its noise contribution is thus imperative in cryo-EM grid preparation. In this paper we introduce MeasureIce, a simple, easy to use ice thickness measurement tool for screening and selecting acquisition areas of cryo-EM grids. We show that it is possible to simulate thickness-image intensity look-up tables, also usable in SerialEM and Leginon, using elementary scattering physics and thereby adapt the tool to any microscope without time consuming experimental calibration. We benchmark our approach using two alternative techniques: the "ice channel" technique and tilt-series tomography. We also demonstrate the utility of ice thickness measurement for selecting holes in gold grids containing an Equine apoferritin sample, achieving a 1.88 Ångstrom resolution in subsequent refinement of the atomic map.

[1] Ian Holmes Imaging Center, Bio21 Molecular Science & Biotechnology Institute, University of Melbourne, Parkville, Victoria, Australia. [2] Department of Biochemistry and Pharmacology and ARC Industrial Transformation Training Center for Cryo-electron Microscopy of Membrane Proteins, The University of Melbourne, Parkville, Victoria, Australia. ✉email: hgbrown@unimelb.edu.au

Single particle cryogenic transmission electron microscopy (cryo-EM) is a characterization technique for biological macromolecules that has seen rapid growth in the past decade[1], both in the number of structures solved and resolutions of the reconstructed 3D molecular models[2]. Sub-3 Å resolutions —which allow the fitting of amino acid side chains in the three-dimensional density map[3]—are now common and the current resolution record is just above 1.2 Å[4]. The history of the development of cryo-EM has principally been one of surmounting the inherent challenges of signal-to-noise. Biological macromolecules are susceptible to electron beam radiolysis, the breaking of chemical bonds by the high-energy electron beam of the transmission electron microscopy (TEM) instrument[5]. Vitrification, plunge-freezing of grids so that the water freezes into amorphous (vitreous) ice, provides some protection by reducing the diffusion of radiolysis products, and so permits stability of at least the low-resolution features of a protein up to an electron dose of around 40–70 e/Å[1,6]. Even with this dose level cryo-EM images are dominated by Poisson counting or "shot" noise, the quantum-statistical fluctuations in the numbers of electrons arriving at any given camera pixel during acquisition, overwhelming the atomic resolution structure that would be otherwise evident in single images. In the single particle analysis (SPA) workflow, sophisticated algorithms[7] combine tens or even hundreds of thousands of images into a single three-dimensional reconstruction such that the available signal then becomes sufficient to finally reach the resolutions approaching what the TEM is inherently capable of.

Since signal to noise is inherently limited in cryo-EM yet critical to successfully solving structures via the technique, much research effort is applied to reducing additional sources of noise. The most striking example is the development of direct electron detectors which feature an improved detector quantum efficiency (DQE), a measure of the additional noise the detection system adds to the measured micrograph[8]. The disordered atomic structure of the vitreous ice that proteins are necessarily encapsulated within in cryo-EM is itself an additional source of noise[9] and thicker ice means noisier images. Therefore, a key step in the cryo-EM single particle analysis (SPA) workflow is the inspection of samples on an economical screening TEM to ensure that sufficient densities of particles can be found in thin ice regions of the grid before these samples advance to data collection on expensive high-end acquisition instruments and consume much needed computational resources. This is hampered by the fact that most methods for quantitative ice thickness measurement

require either time-consuming post-experiment analysis or pre-experimental calibration so are difficult to easily incorporate in screening workflows. Commonly used methods are summarized in Fig. 1 and include tomography, the so-called "ice channel" method, the use of an energy filter or the aperture-limited scattering (ALS) approaches. Tomography, Fig. 1a, has the advantage of providing a maximal amount of information by reconstructing the three-dimensional distribution of protein, including its interaction with the air-water interface, in the ice layer[10]. The longer acquisition and data processing time means it is unsuited to on-the-fly analysis. The ice channel method, Fig. 1b, involves condensing the electron beam to burn a hole through the ice in one stage tilt (e.g., −30 degrees) and then tilting the stage to a second orientation (e.g., +30 degrees) from which the thickness of the ice-layer can be inferred by the projected length of the ice channel in a recorded image[11–13]. In the energy filter approach, Fig. 1c, comparison of the intensities of an un-filtered image and one where a post-specimen energy filter removes electrons that have been inelastically scattered by the ice layer allows for quick on-the-fly measurement of ice thickness[11,13]. The ALS approach, Fig. 1d, uses a post-specimen objective aperture to remove electrons scattered both elastically and inelastically by the ice meaning thicker-ice regions of the grid appear darker in the image[14]. We focus on the ALS method in this work since, once calibrated, ice thickness is measured directly from single micrographs meaning the method is fast, only requires the most basic of screening microscopes without an energy filter or tomography software and minimal user training. In addition, the ALS method is best used at low magnifications where the beam is typically spread so wide that the intensity is much <1 e/Å². Large regions of the grid can be surveyed with minimal damage to the specimen, making this an appropriate technique for selecting holes for acquisition[15], unlike tomography (which requires higher magnification and dose) and the ice channel method which is necessarily destructive to the region of interest.

For ALS images recorded in-focus (so that defocus-induced phase contrast is minimized) the intensity for a region of amorphous ice is well approximated by[14]

$$I = I_0 e^{-t/T_{\text{eff}}}, \quad (1)$$

where $I_0$ is the incident intensity of the electron beam (that measured in the absence of any sample in the path of the beam), t is the thickness of the ice and $T_{\text{eff}}$ is the effective mean-free-path

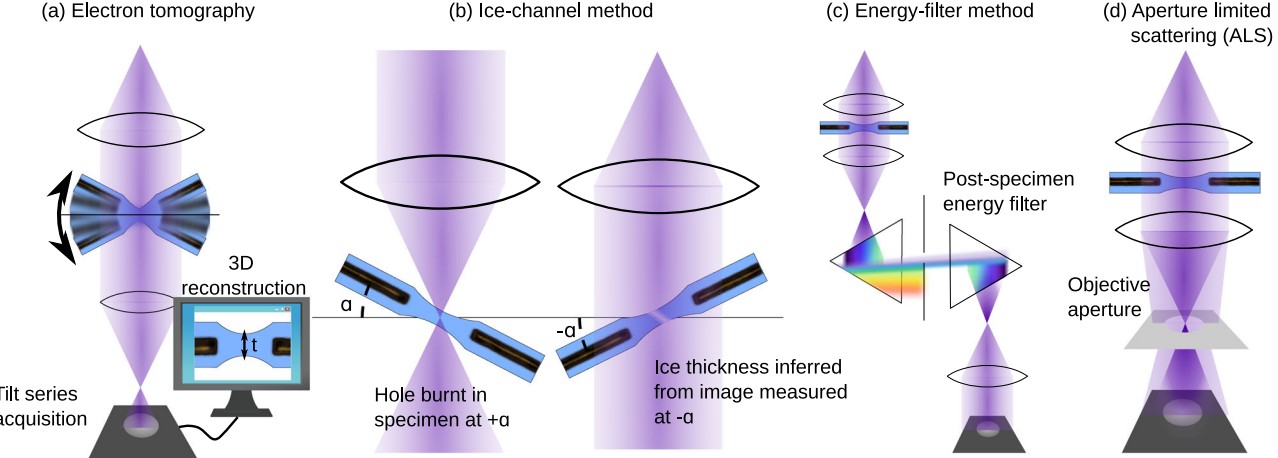

**Fig. 1 Methods for measuring ice thickness in cryo-TEM. a** 3D reconstruction from tilt series tomography, **b** the ice-channel method, where the thickness is inferred by first burning a hole at one sample tilt orientation and then measuring the projected size of the ice channel at another orientation or by measuring the fraction of electron beam blocked by **c** a post-specimen energy filter or **d** objective aperture in the back-focal plane.

for elastic and inelastic scattering of an electron to an angle large enough that the objective aperture will remove it from the image and is often referred to as the ALS coefficient[14]. This effective mean-free-path, $T_{eff}$, is dependent on a complex interplay between elastic and inelastic scattering, which we will review in the methods section of this paper. For a well-aligned microscope it is a function of the accelerating voltage, which determines the degree of elastic and inelastic scattering for a given ice thickness, and aperture size, which determines the scattering angle beyond which scattered electrons will be removed from the image. Note that the relevant value of the aperture size is in units of inverse length or angle (i.e., mrad) measured using a diffraction standard such as gold nanoparticles, not the aperture physical size in micron values typically provided by the instrument manufacturer. In previous work $T_{eff}$ was measured by extensive experimental calibration[13,14]. In this work, we demonstrate how $T_{eff}$ can be estimated through first principles simulation and thus the method be applied to a TEM without the need for the time-consuming calibration step. We benchmark this approach using the ice channel and tomography ice thickness measurements and present a software package named MeasureIce that allows for ice thickness measurements using ALS during data collection. Finally, we demonstrate the utility of quantitative ice thickness measurements pre-acquisition via a 1.88 Å reconstruction of equine apoferritin, which is a 0.22 Å improvement on previously reported results[16,17]. We refer to our technique of measuring ice thickness using aperture limited scattering informed by simulated look-up tables as the "MeasureIce approach".

## Results and discussion

**Comparison with the ice channel approach**. Simulated ice thickness-image intensity curves were generated for three different microscopes operated at different accelerating voltages and are plotted in Fig. 2, a ThermoFisher Talos L120C at 120 kV, Fig. 2a, a Talos Arctica G2 at 200 kV, Fig. 2b, and Titan Krios G4 at 300 kV, Fig. 2c, and using different apertures (indicated in units of microns and mrad, though we stress that it is the latter measure which actually determines the calibration curves). These curves are plotted along with experimental benchmarks using the ice channel method to measure ice thicknesses in regions of vitrified cryo-EM in Fig. 2. Pearson correlation coefficients ($r^2$), indicated on the plots, give the quality of agreement between the MeasureIce simulation calibration curves and experimental datapoints. Good agreement between experiment and theory was found for all experimental accelerating voltages suggesting that the simulations correctly incorporate all of the important and necessary physics for the energy ranges typically used in cryo-EM and can be relied on to produce accurate ice thickness maps.

**Comparisons with tomography**. By way of an additional benchmark, we compared ice thickness measurements from the MeasureIce approach to estimates from tomography. Since information loss in the z direction resulting from the missing wedge in a standard tilt series acquisition makes it difficult to detect the top and bottom surfaces of the ice in a pure water sample, a 7 mg/ml apoferritin sample was used for this experiment. Apo-ferritin particles were used as an indicator of the extent of the water volume.

Four tomograms were acquired using a ThermoFisher Talos Arctica, operated at 200 keV, for different holes in the grid, selected using MeasureIce to give a spectrum of ice thicknesses up to about 100 nm. We then used a custom script to fit a surface function to these points to estimate the profile of the ice layer; this is shown for a representative Quantifoil® hole in Fig. 3a. The ice thickness profile was then calculated as the height difference

between the top and bottom layers of the ice. A comparison of MeasureIce and tomography estimates of ice thickness for all the holes is shown in [Fig. 3c] and reasonable agreement is found between the two approaches. A MeasureIce thickness map incorporating two of the tomograms (1 & 2 or blue and red) is inset in Fig. 3d for reference.

Since de-ionized water was used in the ice channel measurements, using an apo-ferritin sample for this benchmark had the additional benefit of testing whether the presence of a standard cryo-EM protein sample, both protein and buffer, would affect the accuracy of ice thickness measurements. Protein typically has a shorter inelastic mean-free path than the surrounding water so the presence of protein will mean the solution has a slightly shorter inelastic mean-free path on average and MeasureIce might be expected to underestimate thickness. This effect is not observed in our experiments for thicknesses <80 nm though the departure from agreement between the approaches for thicknesses >80 nm might be accounted for by the shorter electron inelastic mean free path of protein-containing solution. This gives confidence in using the MeasureIce approach for standard cryo-EM samples for thicknesses approaching ideal ice thicknesses for SPA acquisition (<50 nm). Caution is advised with samples containing large amounts of additives such as glycerol, or with a concentrated salt buffer.

**Amorphous carbon and graphene-backed grids**. Adding an amorphous carbon or functionalized graphene backing to a cryo-EM grid is becoming an increasingly common practice, since this sometimes addresses preferred orientation effects[16] and the conductive film ameliorates beam induced charging and thus ice movement[18]. Since a graphene backing should be only a few monolayers thick (therefore less than a nanometer) we expect this to have negligible effect on ice thickness measurement. Amorphous carbon backings are usually thicker, typically 5–10 nm, so will impact ice thickness measurements. Simulations we present here demonstrate that accounting for a carbon backing in ice thickness measurements is straight-forward and does not even require a specialized calibration for different thicknesses of carbon film. All that is required is that the reference intensity, $I_0$, instead be set to the measured intensity for the carbon film no ice (a dry region of the backed grid). Then a standard $I/I_0$ calibration curve prepared for a setup assuming a non-carbon-backed grid can be used to measure ice thickness. A 10 nm amorphous carbon layer was added to the MeasureIce simulation for a 200 kV microscope using a realistic carbon atomic model generated by Ricolleau et al.[19] and an inelastic mean free path for an electron in carbon from Shinotsuka et al.[20]. As can be seen when the fractional intensity $I/I_0$ is plotted against ice thickness for two different hypothetical objective apertures, 10 mrad and 15 mrad, in Fig. 4, the effect of the carbon film is to shift the y-axis intercept of the line but not affect the slope of the line appreciably. This is confirmed by the ALS coefficients, which for the most part change by <1 nm. Adjusting $I_0$ to the intensity measured for just the carbon film, that is for an ice thickness of 0 nm, would then shift any of these calibration curves back to that observed for no carbon film.

**Uncertainty in literature values of inelastic mean free path**. Energy-dependent inelastic mean free path, $T_{imfp}$, is a key parameter in MeasureIce simulations and its accuracy will have consequences for ice thickness measurement. In this section, we will discuss the considerable variation in literature values of inelastic mean free path but show that the possible impact of this variation on the estimated ice thickness is of the order 10%. The mean free paths for inelastic scattering, $T_{imfp}$, which are stored

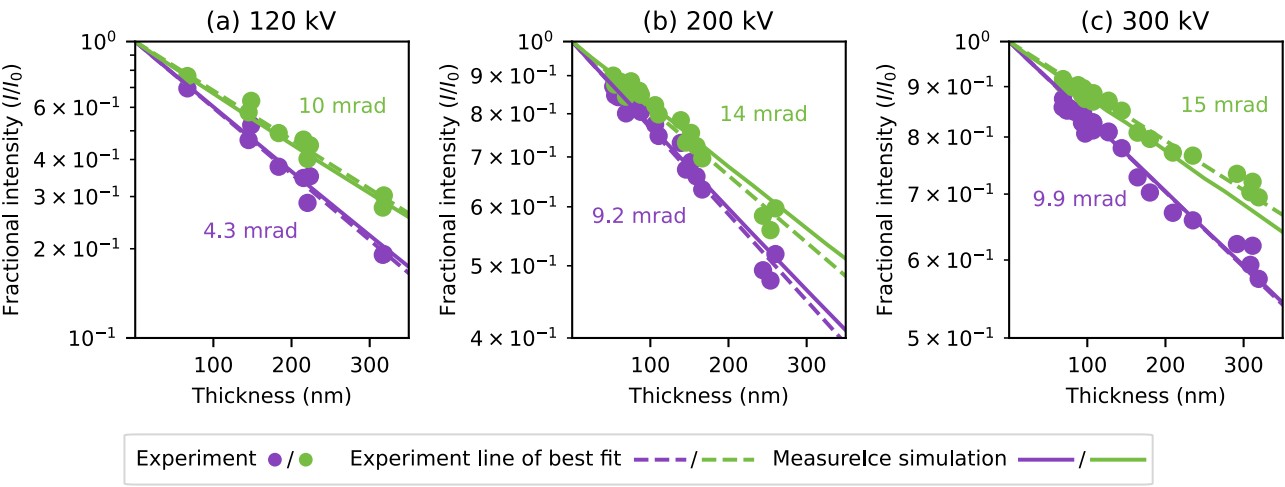

**Fig. 2 Simulated Ice thickness—image intensity maps (MeasureIce simulation) for three different microscope accelerating voltages.** Displayed are **a** 120 kV, **b** 200 kV, and **c** 300 kV and different objective apertures with benchmarking using the ice channel method (Experiment) along with the line of best fit. The Pearson correlation coefficients ($r^2$) indicate the quality of agreement between experimental datapoints and the MeasureIce simulation (not the experiment line of best fit).

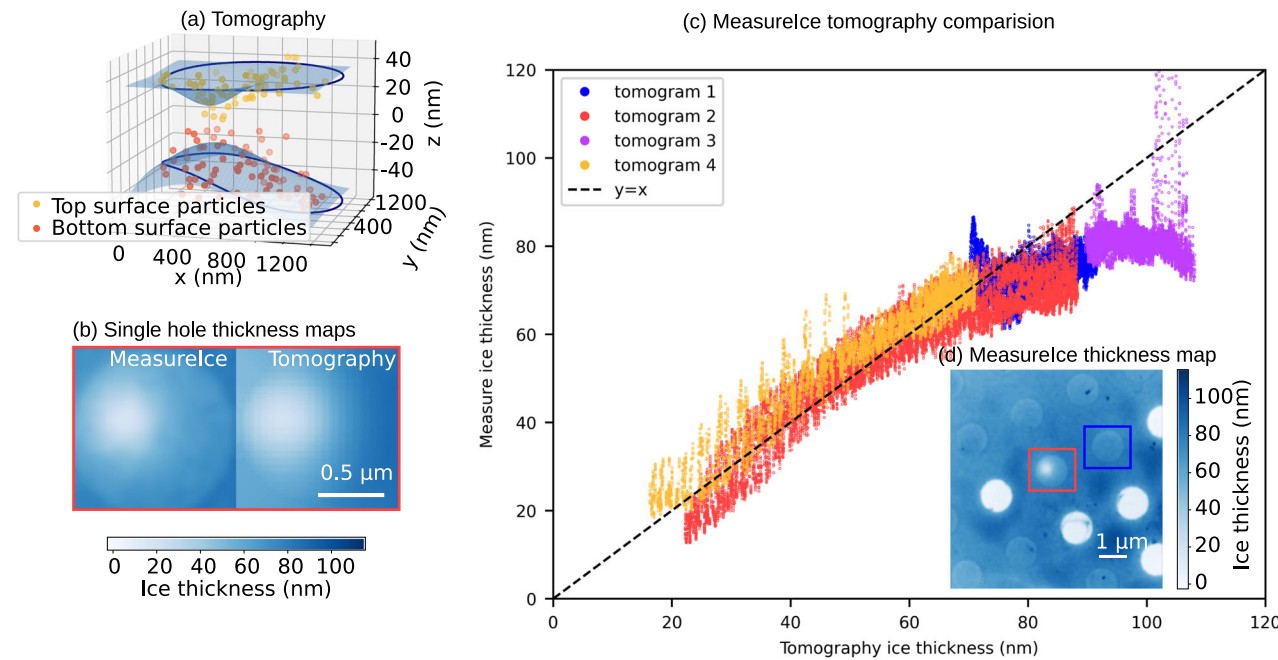

**Fig. 3 Benchmarking of MeasureIce with electron tomography.** For each hole particles at the extremities in z were manually picked from the grid and a surface fitted to these points (**a**), the height difference between these two surfaces is taken to be the estimated thickness and this is plotted in (**b**) alongside the same grid hole measured using the MeasureIce approach. Shown in (**c**) is a comparison of ice thickness as measured by tomography and MeasureIce for four different grid holes on an apo-ferritin containing carbon Quantifoil grid. Each datapoint in (**c**) is a single pixel in the MeasureIce thickness map. The MeasureIce thickness map for two of the tomograms is inset in (**d**).

internally in MeasureIce, are taken from recent measurements by Yesibolati et al.[21] for 120 kV and 300 kV. For other values electron energy we use the following equation[22,23] as a basis for interpolation between measured values of T[21]

$$T = \frac{M_W \beta^2}{9.03 \times 10^{-10} \rho Z^{1/2} \ln \frac{\beta^2 (U_0 + mc^2)}{10}}. \qquad (2)$$

Here $M_w$, $\rho$, and $Z$, are the molecular weight, density, and atomic number of the sample and $U_0$, $m$, $c$, and $\beta$ are the beam energy, electron mass, speed of light, and electron speed (as a fraction of c). Experimentally measured values for the inelastic mean free

path[21,23–26] and those predicted by Eq. (2) and our fit of Eq. (2) to the results of Yesibolati et al.[21] (labeled "Eq. 2 adjusted") are plotted in Fig. 5. We consider the result of the experiment of Yesibolati et al.[21] the most rigorous to date since mean free path is directly measured from electron energy-loss spectroscopy (EELS) data recorded for an electron probe scanned across water channels of known geometry. Grimm et al.[24], whose value for the inelastic mean free path of an electron at 120 kV agrees well with Yesibolati et al.[21], conducted a rigorous error analysis with their experiment and this presents an opportunity to investigate the effect of errors in inelastic mean free path. Figure 5b is the same as Fig. 2a, which compares MeasureIce simulations with

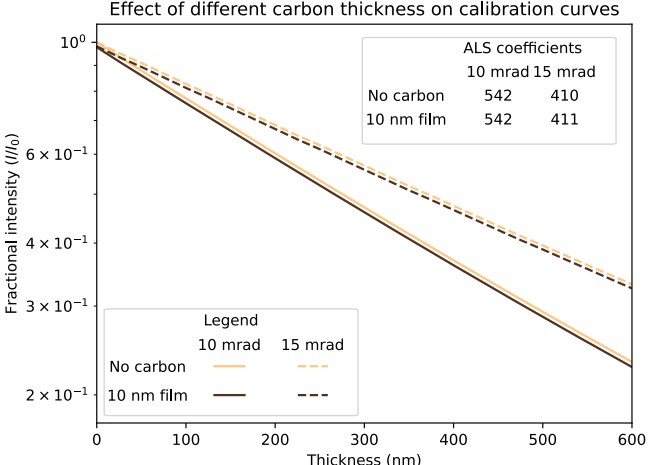

**Fig. 4 MeasureIce generated calibration curves with and without a 10 nm amorphous carbon cryo-EM grid backing.** The curves are linear if fractional intensity is plotted with a log scale and the effect of the carbon backing is to shift the origin of the line but not its slope (the ALS coefficient) so only the reference intensity ($I_0$) need be set to that measured for just a carbon film for ice thickness measurement with carbon backed grids.

benchmark ice thickness measurements, but with additional MeasureIce calibration curves, generated with the inelastic mean free path changed to the extremities of Grimm et al.'s[24] error bars and plotted with dashed lines. Inset in this figure is a plot of the implied thickness error for both the L120C apertures as a function of nominal thickness (i.e., Implied by the Yesibolati et al.[21] value of the inelastic mean free path). It can be seen that this implied ice thickness error is consistently well below 10%, sufficient for estimating ice thickness in cryo-EM.

**MeasureIce, a software tool for measuring ice thickness**. Once ice thickness image intensity look-up tables are simulated for a microscope's given aperture-accelerating voltage combination this information can be used to infer ice thickness on the fly. We have developed a Python software tool to facilitate thickness measurements using these simulated look-up tables using the pyqtgraph (https://www.pyqtgraph.org/) graphics and user interface Python library, though stress that the code outputs values of $T_{eff}$ that can be used in existing software, notably Leginon[14] and SerialEM[15]. The source code for the MeasureIce python script is available on GitHub (https://github.com/HamishGBrown/MeasureIce). It can be deployed using a python interpreter installed on the microscope control or support PC or compiled into a stand-alone Windows executable using the py_installer (https://www.pyinstaller.org/) package.

Ideally, Cryo-TEM images for ice measurement are recorded with the specimen in focus and using parallel illumination with a well-aligned objective aperture positioned in the back focal plane, however, our own tests (not shown) suggest that deviating from parallel illumination induces minimal error.

A screenshot of the graphical user interface is shown in Fig. 6. The raw TEM image is shown in the left panel, Fig. 6a. The user can select between different pre-computed image intensity-thickness calibrations (which are stored as hdf5 files, https://www.hdfgroup.org/HDF5/), for example for different microscopes or for different accelerating voltages on the same microscope, from the drop-down menu at the bottom left, Fig. 6b. Raw images are loaded as tiff, ser (using the openNCEM, https://github.com/ercius/openNCEM, package developed by Peter Ercius at Lawrence Berkeley National Lab) or mrc [using the Python mrcfile module[27]] files by clicking the "Load raw image"

button, Fig. 6c. The quantity of interest is the beam intensity as a fraction of the incident intensity ($I/I_0$), the image intensity recorded for regions of the image with no material in the path of the beam, which we will refer to as the "vacuum intensity". MeasureIce needs the user to manually input $I_0$ and this is measured from a region of the image without ice or TEM grid overage. The vacuum intensity $I_0$ can be selected by moving with the mouse pointer the red line, labeled $I_0$, which is atop the histogram immediately to the right of the raw image, Fig. 6d, to the appropriate value on the histogram. Regions of the image with intensity $>I_0$ are highlighted in red in the interface, Fig. 6a. Once $I_0$ has been set, the user generates an ice thickness map by pressing the "Generate Ice thickness map" button, Fig. 6e. Ice thickness measurements can be read off the interface by hovering the mouse pointer over the region of interest in the ice thickness map, Fig. 6f. The ice thickness map can be saved as a 32-bit tiff file for reference and further analysis or as a matplotlib[28] plot in png or pdf format by clicking the "save ice thickness map" button, Fig. 6g. More detailed step-by-step instructions for MeasureIce may be found in the "Methods" section.

**Using MeasureIce for screening and data collection**. The ultimate goal of MeasureIce is to ensure that cryo-EM practitioners can more rapidly and quantitatively achieve minimum possible ice thicknesses for a given sample, which in turn reduces the effective signal to noise of micrographs resulting in higher resolution reconstructions, potentially from fewer total particles. To this end MeasureIce was evaluated on a standard cryo-EM single particle workflow with Equine Apoferritin (Sigma) vitrified on UltrAuFoil® gold grids.

Compared to carbon Quantifoils® where ice thickness gradients are visible even on the carbon substrate and give some qualitative indication of ice thickness across a grid square, gold grids can be challenging for qualitatively evaluating ice thickness for inexperienced users. This is because the gold foil scatters the electron beam strongly and Bragg scattering by the polycrystalline foil dominates TEM images outside the grid holes, rendering the subtle ice thickness gradients invisible. Using MeasureIce on the microscope support PC we were able to identify the relative intensity that corresponded to ice holes having <15 nm thickness and select these holes for acquisition using the "Ice filter" setting on the EPU automated acquisition software of our Titan Krios—which selects holes for acquisition based on the mean image intensity inside the hole. Data processing was performed with Relion[7] and Cryosparc[29] and resulted in a 1.88 Å map. This exceeds the current record resolution 2.1 Å map of the same protein[16,17]. The reconstructed map is shown overlaid with the fitted molecular model 6RJH from ref. [1] in Fig. 7a with a representative micrograph shown in Fig. 7b. The gold-standard Fourier shell correlation curve (GSFSC) is shown in Fig. 7c along with a rendering of the total EM map. Given differences in microscopes, cameras, grids, and reconstruction workflows, we do not seek to attribute wholly this improvement to quantitative ice thickness curation but suspect that thin ice was a key factor. We also note that in other work, thinner ice has been shown to lead to higher resolution 3D maps for symmetric particles, but thinner ice can lead to a worsening of preferred orientation that can degrade the reconstruction resolution for asymmetric particles[30].

**Conclusions**. The aperture limited scattering (ALS) approach provides a fast and convenient way to measure ice thickness on the fly, though till now the time-consuming nature of calibration means that the method is not as widespread in TEM as might ideally be the case. In this paper we have presented an approach to calculate look-up tables to convert image intensities in the ALS

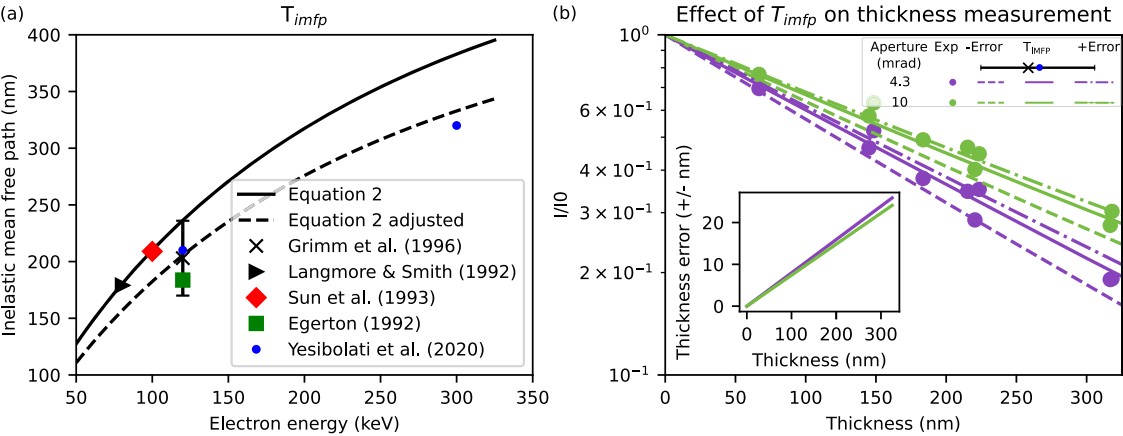

**Fig. 5 Variance in measurements of inelastic mean free path and its impacts on MeasureIce. a** Literature values for experimental mean free path plotted alongside Eq. 2, and the interpolating function for mean free paths used in MeasureIce (Eq. 2 adjusted). **b** Ice thickness measurement errors that would be implied by the effect of experimental errors from Grimm et al.[24]. Solid line is the calibration curve implied by Yesibolati's mean free path measurement at 120 kV[21] and the dashed lines represent the extremities of the error bars in (**a**), the plot inset is the error as a function of thickness implied by Yesibolati's measurements.

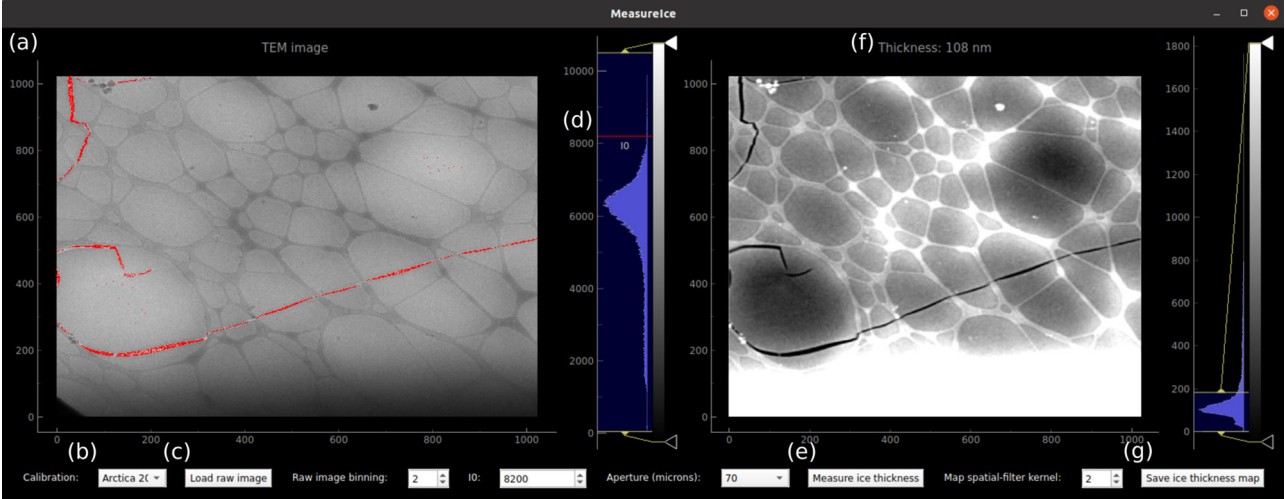

**Fig. 6 Screenshot of the MeasureIce software tool. a** In the left panel a raw TEM image is displayed, **b** the calibration file is selected via the drop-down menu, and **c** a new image can be loaded by pressing "Load raw image". **d** The quantity $I_0$, the incident intensity of the electron beam is set using the red line on the histogram or using the input on the bottom of the interface. **e** An ice thickness map is generated by pressing the "Measure ice thickness" button with (**f**) the ice thickness map is displayed in the right panel of the interface which can be saved by pressing (**g**) "Save ice thickness map".

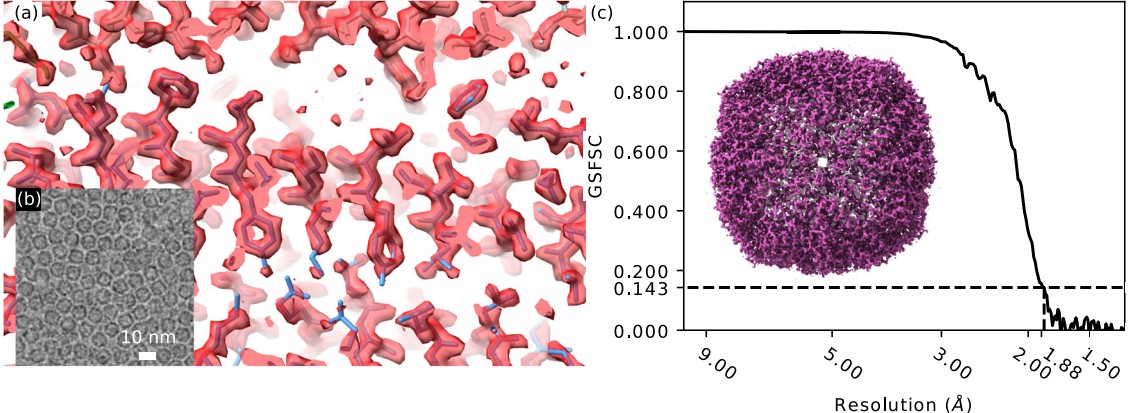

**Fig. 7 MeasureIce assisted single particle reconstruction of Equine apoferritin. a** Comparison of Cryo-EM reconstruction with structure model from Naydenova et al.[1], **b** a representative micrograph, with a 1 nm low-pass filter, and **c** gold-standard Fourier shell correlation (GSFSC) assessment of map resolution with a rendering of the reconstructed molecular map inset.

approach to ice-thickness values that allows the time-consuming process of experimental calibration to be skipped. This approach has been packaged for user convenience in software which we have named MeasureIce meaning that barriers to including quantitative ice thickness measurements are substantially reduced. Thickness measurements compare favorably with alternative ice thickness measurement techniques, and we have demonstrated the advantages of quantitative ice thickness measurements in general with a 1.88 Å resolution reconstruction of Equine apoferritin. This improvement cannot wholly be attributed to ice thickness measurement but does, however, make the case for including quantitative ice thickness tools like MeasureIce within cryo-EM screening and acquisition workflows, especially considering the convenience of the MeasureIce software tool.

## Methods

**Simulating ice thickness maps.** In simulations, we found that it was important to include both elastic and inelastic scattering and the realistic structure of amorphous ice. A 10 nm³ cube of vitreous ice was generated using the GROMACS molecular dynamics simulation package[31] according to the method detailed in Souza Junior et al.[32]. The multislice algorithm has been previously used to simulate electron elastic scattering in the cryo-EM context[22,33,34] and here simulations were performed using the open source py_multislice package[35]. To incorporate multiple inelastic scattering we adapted the approach suggested by Egerton[36]. For a given thickness $t$ the probability of an electron being inelastically scattered $n$ times due to plasmon excitation is given by,

$$P_n(t) = \frac{I_n(t)}{I_t} = \frac{1}{n!}\left(\frac{t}{T}\right)^n e^{\frac{-t}{T}}. \qquad (3)$$

The differential cross section for inelastic scattering to angle $\theta = \tan^{-1}(\lambda k)$, where $k$ is the diffraction plane coordinate in units of inverse length and $\lambda$ is the electron wavelength, is parameterized as

$$\frac{d\sigma}{d\theta} = \frac{1}{\theta^2 + \theta_E^2}, \qquad (4)$$

where $\theta_E$ is the energy-dependent characteristic scattering angle for plasmon excitation, this is provided by Yesibolat et al.[21] for 120 kV and 300 kV and interpolated for other values. Since $\theta_E$ is typically a fraction of mrad it tends to mainly determine the scattering distribution at small angles (<1 mrad), even doubling this figure in an ad hoc way was not found to noticeably affect the results of interest since most apertures are >5 mrad. To incorporate plasmon inelastic scattering into simulation the electron wave function intensity resulting from the multislice simulations are convolved with the differential cross section in reciprocal space $n$ times for each instance of scattering and weighted with the probability that the electron will scatter that number of times from Eq. (3),

$$\left|\Psi(k,t)\right|^2 = P_0(t)\left|\Psi_0(k)\right|^2 + P_1(t)\frac{d\sigma}{d\theta}*\left|\Psi_0(k)\right|^2 + P_2(t)\frac{d\sigma}{d\theta}*\frac{d\sigma}{d\theta}*\left|\Psi_0(k)\right|^2\ldots \qquad (5)$$

The image intensity, as a fraction of incident beam flux $I_0$, for a given thickness of ice $t$ and acceptance angle $\alpha = \tan^{-1}(\lambda k_\alpha)$ of the objective aperture, is then given by

$$I(t)/I_0 = \int_0^{k_\alpha}\left|\Psi(k,t)\right|^2 dk. \qquad (6)$$

Fractional image intensity as a function of ice thickness is calculated by the tool Generate_MeasureIce_calibration.py and the user need only provide electron energy in keV and apertures in units of inverse Ångstrom or mrad to this utility. The script will produce calibration files for the stand-alone MeasureIce graphical user interface and also report $T_{\text{eff}}$, the ALS coefficient, to the user so that it can be used in the Leginon[14] and SerialEM software[15].

**Experimental benchmarking of the ice thickness look-up tables.** To confirm the validity of the simulation approach we benchmark it using the ice channel approach. Here the sample is tilted to angle $\alpha$ and the beam is focused to a small spot, see Fig. 1a, to burn an ice channel. Next, an image is recorded of the ice channel with the stage tilted to angle $\beta$ in the opposite direction. The thickness of the ice layer can be inferred from the size of the ice channel measured in projection,

$$d = t\frac{\sin\alpha + \beta}{\cos\alpha}. \qquad (7)$$

It is most convenient to choose $\alpha = \beta$, i.e., symmetric tilts, since then the projected size of the entrance and exit holes of the ice channel in the image will remain unchanged between the two acquisitions and the above equation simplifies to

$$d = 2t\cdot\sin\alpha. \qquad (8)$$

The process is also considerably simplified by acquiring images at both tilts with a

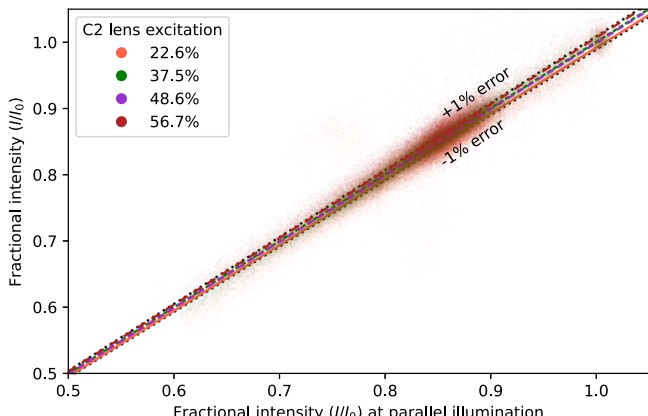

**Fig. 8 An exploration of the errors introduced in the measurement of TEM image fractional intensity with non-parallel illumination.** For an ice-covered carbon Quantifoil TEM grid, images were recorded for different values of C2 excitation, which cause non-parallel illumination if not set to 48.6% at the chosen spot size. Pixel values are compared with their equivalent in the image recorded with parallel illumination. Even a large change to the C2 excitation (22.6%) is not enough to get a >1% error in measured fractional intensities.

large overfocus (around 20 μm) since this has the effect of high-pass filtering the images, making the edges of the ice channel entrance and exit holes more recognizable in the raw images.

**Recording images for MeasureIce on the TEM.** MeasureIce images must be recorded in focus and with the objective aperture centered with respect to the TEM optic axis. MeasureIce simulations assume that the beam is in a parallel illumination condition [see e.g., ref. [37]] and the objective aperture is accurately placed in the back focal plane. Own tests which we discuss in the following section indicate that measuring ice thickness with the beam strongly deviated from this condition should only introduce imperceptible errors. The reference intensity $I_0$ also needs to be known accurately. If there is no hole in the specimen through which $I_0$ can be estimated, then a separate reference image, recorded with the same beam intensity, magnification, and acquisition time should be recorded to measure $I_0$.

**Minimal effects on thickness measurement by deviation from parallel illumination.** To explore the effect of non-parallel illumination, we compared the fractional intensity measured in an image of a carbon quantifoil grid with vitreous ice layer recorded at a parallel illumination condition in our Arctica G2 cryo-TEM (a C2 lens excitation of 48.6% at spot size 5) with TEM images of the same area but with different values of the C2 lens excitation, causing some departure from the parallel condition. The 70 micron (9.2 mrad) objective aperture was inserted. Per pixel values are plotted in Fig. 8 with a line of best fit for each value of C2 lens excitation. Even a very large departure from parallel (22.6% C2 excitation) was not sufficient to get an ~1% difference in measured fractional intensity with respect to the parallel condition indicating that errors introduced in ice thickness measurement by non-parallel illumination are likely to be small. For this instrument and aperture combination the most extreme values of C2 excitation would cause a thickness error of <4 nm for ice thickness values <50 nm (the target for single particle cryo-EM), much larger sources of error are considered elsewhere in the paper.

**MeasureIce instructions.**

1. Select the appropriate calibration file from the bottom left-most drop-down menu. These must be generated beforehand using the Generate_measureice_calibration.py utility.
2. Load a TEM raw image (mrc, tiff, or ser) using the "Load raw image" button. Users must be cautious that microscope software is not compressing the bit depth of images. For example, saving with the tiff format in ThermoFisher's TIA software compresses to the 8-bit or 16-bit integer datatype. Bit depth compression will mean intensity measurements are not comparable between acquisitions even if beam current and camera acquisition time is kept constant. To ensure accurate ice thickness measurements, users must be also careful that the image was recorded in focus (with minimum defocus induced phase contrast) and with the objective aperture aligned to be concentric with the microscope optic axis. The use of the "counting" mode for direct electron detectors such as the Gatan K2 and K3 or ThermoFisher Falcon cameras is discouraged since

**Table 1 Comparison of apoferritin data collection and reconstruction parameters between this paper and that in Naydenova et al.[1].**

|  | This paper | Naydenova et al.[16] |
|---|---|---|
| Grids | Gold Ultrafoil 1.2/1.3 | Functionalized graphene-coated gold Ultrafoil 1.2/1.3 |
| Microscope | Titan Krios G4 | Titan Krios |
| Accelerating voltage (kV) | 300 | 300 |
| Magnification | 120,000 | 120,000 |
| Pixel size (Å) | 0.637 | 0.6495 |
| Exposure time (s) | 1.52 | 30.0 |
| Dose (e-/Å$^2$) | 16 | 37 |
| Number of Particles used in final refinement | 68,025 | 41,202 |
| Gold standard FSC map resolution (Å) | 1.88 | 2.14 |
| B-factor (Å$^2$) | −54 | −54 |

exceeding the recommended dose-per-pixel for accurate electron counting will mean a high rate of electron coincidence loss and that the detector response will not be linear. We have observed that the MeasureIce software noticeably underestimates ice thickness values when this happens. The low magnifications at ice thickness maps that will normally be recorded (typically around 5000 times magnification) mean that the dose efficiency improvements resulting from electron counting are anyway irrelevant since the dose applied to the specimen should be a fraction of an electron/Å$^2$.

3. Set the vacuum intensity $I_0$, the image intensity recorded for regions of the image with no material in the path of the beam, by moving with the mouse pointer the red line superimposed on the intensity histogram immediately to the right of the raw image. The MeasureIce interface will identify regions of intensity greater than this value with a red mask overlayed on the image. This is shown for the case of a crack in the holey carbon film visible in the TEM image of Fig. 5. For images of grid regions with complete ice coverage $I_0$ needs to be measured from a separate image of an empty part of the grid using identical beam current and acquisition time and manually inputted into the $I_0$ section of the interface.

4. Select the aperture used to record the image from the drop-down menu in the center of the bottom panel of the interface—MeasureIce needs the aperture sizes in diffraction units of inverse Angstrom or mrad but the apertures can be labeled with the micron values that will actually appear in the microscope user interface.

5. Press the "Measure ice thickness" button. The ice thickness map will be shown in the right-hand panel of the user interface and hovering the mouse above regions of the ice thickness map will cause the ice thickness for the region below the cursor to be displayed in the image title. Since there can be apparent large pixel-to-pixel variation in the ice thickness map, due mainly to Poisson counting noise, there is the option of filtering the image to remove much of this noise using a Gaussian low-pass filter. The amount of filtering is controlled by setting the filter kernel size in the "Map spatial-filter kernel" radio box.

6. Save the ice thickness map for future analysis and presentation by clicking the "Save ice thickness map" button.

**Grid preparation**. Equine Apoferritin at 7 mg/ml (Sigma, A3641) was prepared on 1.2/1.3 gold Ultrafoil grids, glow discharged in a PELCO easiGlow system for 2 min with a 15 mA current, for single particle reconstructions and on 1.2/1.3 carbon Quantifoil grids, glow discharged for 30 s with a 15 mA current, for tomography using a Leica EM GP2 automated plunge freezer. A 4 µL volume was applied to the grids which were back-side blotted using the 'auto-blot' function of the EM GP2. The sample chamber was set to 22° and 95% humidity.

**Tomography acquisition and reconstruction**. Images were recorded on a Talos Arctica (FEI) fitted with a Gatan K2 camera operated in electron counting mode and energy filtered with a 20 eV slit inserted to improve contrast. Before tilt-series acquisition, low magnification (×5000) images were recorded for later MeasureIce analysis. The tomograms were reconstructed using IMOD[38], with frames aligned via cross-correlation and binning of 4 K images by a factor 4 to enhance the contrast of the apo-ferritin particles. Particles at the extremities (in z) of the reconstructed volume were taken to indicate the upper and lower layers of ice and manually picked.

**Single particle analysis acquisition and reconstruction**. Grids were imaged on a Krios G4 cryoTEM (ThermoFisher) equipped with a Falcon 4 direct electron

detector. The fractional intensity implied by MeasureIce for ice thickness <15 nm was used to select holes in the ThermoFisher EPU software using the "Ice filter" feature. The samples were imaged at a nominal magnification of 120,000 corresponding to a pixel size of 0.637 Å, calibrated using the 220 lattice reflection of a standard orientated gold specimen, with a defocus of −0.4 µm to −1.0 µm. Exposures of 1.52 s were taken with a fluency of 5.34 e/px/s resulting in a nominal dose of ~20 e-/Å$^2$ on the sample with 8 frames per movie. The last frame was deleted to avoid unevenness of the dose distribution resulting in 7 frame movies with a total dose of ~16 e/Å$^2$. Acquisition parameters are compared with those of Naydenova et al.[16] in Table 1.

Reconstruction was carried out using a mixed workflow of RELION 3.1[8] and Cryosparc 3.0[29]. Motion correction was performed on gain-corrected movies with RELION 3.1's own motion correction implementation. Motion corrected movies were imported into Cryosparc with CTF estimation done using patch CTF estimation. A total of 97,513 particles were extracted. Two rounds of 2D classification in Cryosparc resulted in a working dataset of 68,025 particles. Particles were imported back into Relion using Pyem[39]. Particles were used to create a de novo model then subjected to 3D refinement, CTF refinement, and polishing follow by another round of CTF refinement. Octahedral symmetry was enforced on the 3D model in 3D refinement. The ultimate 3D refinement resulted in a structure at 1.9 Å with a B-factor of −66. Ewald sphere correction was applied to the refinement results and improved the structure to a resolution of 1.88 Å and a B-factor of −54. The resolution was estimated using the gold standard FSC = 0.143, calculated using a relaxed solvent map (deposition EMD-25619).

**Model fitting**. Pdb structure 6RJH[16] was fitted into the Coulomb potential map using rigid body fitting in Chimera-X[40].

**Reporting summary**. Further information on research design is available in the Nature Research Reporting Summary linked to this article.

## Data availability
The apoferritin map is available on the electron microscopy databank (EMD-25619) and the original unaligned movies are available on the EMPIAR database (EMPIAR-11013).

## Code availability
The Python source code for MeasureIce (https://github.com/HamishGBrown/MeasureIce, https://doi.org/10.5281/zenodo.5751190) and py_multislice[35] (https://github.com/HamishGBrown/py_multislice, https://doi.org/10.5281/zenodo.5762736) are available via GitHub.

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

## Acknowledgements
We acknowledge use of TEMs and sample preparation equipment at the Ian Holmes Imaging center at the Bio21 Institute of the University of Melbourne. H.G.B. acknowledges support through the University of Melbourne Early Career Researcher (ECR) Grant Scheme. Dr. Ann Frazier is acknowledged for final proof reading of the manuscript.

## Author contributions
H.G.B. and E.H. both conceived the research project and H.G.B. developed the software and underlying theory of MeasureIce and performed the validation experiments under the supervision of E.H. H.G.B. recorded the apo-ferritin dataset and both H.G.B. and E.H. performed the data processing. H.G.B. wrote the manuscript with reviewing and editing assistance from E.H.

## Competing interests
The authors declare no competing interests.
