## [Peer Review File · Communications Biology]

Reviewers' comments:

Reviewer #1 and co-reviewer (Remarks to the Author):

To: Manidipa Banerjee, PhD, Editorial Board Member, Communications Biology

RE: MeasureIce: Accessible on-the-fly measurement of ice thickness in cryo-electron microscopy

Having appropriate ice thickness is important for achieving maximal resolution in cryo-EM. Finding and selecting the appropriate areas in the grid during TEM imaging is also important, as this determines the efficiency of the rather costly data collection sessions. In this work, Brown & Hanssen develop a new tool, MeasureIce, that simulates ice thicknesses from first principles, and can be used in the screening and selection of regions of cryo-EM grids with suitable ice.

The major strength of this work is the authors' use of elementary scattering physics to simulate ice thicknesses in cryo-EM images. They benchmark their simulations using both ice channel and tomography methods, which are established methods for determining ice thicknesses, and find good agreement with their simulations. This is an elegant approach, and as the authors mention, bypasses the need to experimentally calibrate each microscope, which can be time consuming. Finally, a software tool, MeasureIce, is developed for visualizing ice thicknesses. Overall, the paper is well considered, meticulous, and accessible, and should provide another tool for cryo-EM scientists to estimate ice thicknesses. I recommend accepting with minor revisions.

I have the following comments and questions:

1. Since energy filters are widely in use on many Krioses, can the authors comment on the applicability of their simulations to such systems? In Rice et al 2018 the ALS coefficients for a Krios with and without an energy filter and at different slit widths are quite different, so one might expect this to influence the simulated ice thicknesses?
2. In Figure 3 the MeasureIce vs tomography ice thickness starts to deviate from ideal at > 80 nm. Why is this?
3. Is MeasureIce integrated with the EPU workflow? Or is it check out actual ice thickness values with MeasureIce ◊ Compare holes in MeasureIce to EPU ◊ set EPU ice filters. The way MeasureIce is used for "on-the-fly" screening was not immediately clear to me.
4. The authors recommend hole-magnification images to be taken at close to focus to reduce defocus-induced phase contrast. In practice, these images are routinely taken at ~-50 μm defocus for better contrast for hole finding. How do the authors balance the need for defocus-induced contrast for hole finding with the need to be close to focus for accurate ice thickness measurements?
5. Under the section "Uncertainty in literature values of inelastic mean free path", there is this sentence: "However it must be acknowledged that the inputted value of inelastic mean-free path is the most obvious source of error in the MeasureIce calculations." It wasn't quite clear to me what this meant, as inelastic mean-free path does not appear to be a direct input option in MeasureIce.
6. It took me several tries to digest the section "Uncertainty in literature values of inelastic mean free path". If the authors would write a sentence or two at the start of that section to summarize the problem, I think that would help a lot with setting the context.
7. In my opinion the authors should move the final sentence in "Using MeasureIce for screening and data collection" into "Conclusions". The sentence reads: "This result does however make the case for including quantitative ice thickness tools like MeasureIce within cryo-EM screening and acquisition workflows, especially considering the convenience of the MeasureIce software tool." A 1.88 Å reconstruction of apoferritin on its own, without the appropriate controls, does not make a case for ice thickness curation. However, the case is made when considering the story in its entirety and in context of the existing literature, which the authors have thoroughly explained in the full text.

And a few additional comments:

- It is odd to describe SPA as a characterization tool. Also perhaps refer readers to a nice recent review in the first lines of the Introduction section. That might be more helpful than other somewhat random details provide (e.g. the dose tolerance from Grant and Grigorieff which I think is no longer the accepted norm). Similarly to quote a paper from 2012 describing resolution capabilities seems inappropriate. As is the term "recent" for describing direct detectors.
- It would have been interesting to see a comparison of the results of the MeasureIce method to those obtained from an energy filter using the Method described by Rice et al.
- Could the authors comment on the disparity between the ice thickness ranges that are measure for the tomography tests (between 20 and 100nm) vs. the hole drilling test (much thicker and few below 100nm).
- I do not think that 2.1 Å is any longer the best available resolution for Apof.

Reviewer #2 (Remarks to the Author):

Brief summary of the manuscript:

Over the last 10 years single particle cryo-electron microscopy (cryo-EM) has become a mainstay technique in structural biology and this method has been used to solve several high resolution structures. Despite the fantastic progress made to streamline cryo-EM data collection and subsequent image processing and 3-D reconstructions, several bottlenecks remain. One of the major bottle is identification of which areas of a vitrified grid will be suitable for data collection for high resolution structure determination and this step often requires the input of expert electron microscopists. Whether micrographs have been collected from suitable areas of the grid often becomes apparent after the data collection session and this may lead to wastage of valuable microscope and user time. One of the major consideration for identifying "good" areas on a grid is the optimal thickness of the vitreous ice. If the ice is too thin then particles will not be embedded in the ice and too thick an ice layer will contribute to added background noise. In this work the authors have formulated a standalone ice thickness determination software called MeasureIce. This software can be used on all microscopes and does not require any experimental calibration. Using Apoferritin as a test sample, the authors demonstrate MeasureIce guided selection of grid holes can lead to high resolution reconstruction.

Overall impression of the work:

This work has the potential to streamline cryo-EM data collection by introducing the objective metric of ice thickness as a guide for selecting good areas of a grid. Moreover by reducing the dependence on an expert electron microscopist for data collection, MeasureIce might aid in further democratization of cryo-EM. The ease of adapting MeasureIce to different electron microscopes hints at the wide use and versatility of this tool. However, a few experiments and clarifications (detailed below) would increase the impact of the work.

Major comments:

- 1) According to Rheinberger et al. the apparent mean free path (T_{eff}) depends on a number of factors including the magnification regime of the microscope (for instance LM, M or SA mode for a FEI/TFS microscope). If that is indeed the case then it must be mentioned what magnification regime is optimal for using MeasureIce and if T_{eff} is not dependent on the magnification then an explanation should be provided about the discrepancy between the author's assumption and the previous literature.
- 2) A detailed description of how MeasureIce was used during a data collection session would be useful for microscopists planning to use the software. The following information will be useful for potential users:

- a. Whether there is any particular microscope alignment (other than the direct alignments performed routinely) that is crucial for the success of MeasureIce
- b. At which stage ice thickness was determined (while selecting squares from grid atlas or selecting holes in a grid square or selecting exposure area in a hole)
- c. Was MeasureIce used once during the data collection session to figure out the intensity values corresponding different ice thicknesses or was ice thickness measured at multiple points (for instance while selecting holes within each grid square)
- d. What magnification, beam condition etc. were used for acquiring the image that acted as the input for MeasureIce

3) As has mentioned by the authors, the high resolution reconstruction of the Apoferritin test sample might not be solely due to the use of MeasureIce. It would be useful to clearly demonstrate whether ice thickness can indeed be used as an objective metric for collecting high quality micrographs. Using MeasureIce the authors might collect images with optimal ice condition (<15 nm) and with different suboptimal thicknesses of ice (for instance 30 nm, 50 nm etc). 3D Reconstructions calculated with particles from different ice thicknesses might then be compared (in terms of resolution, B-factor etc.) to determine whether ice thickness can be used as the sole metric for selecting suitable areas of a grid.

Minor comments:

1) It will be useful to comment on whether MeasureIce can be used for grids having a support layer of carbon or graphene over the holes.

2) In page2 the authors mention that Bayesian algorithms are used for image processing. This is not completely accurate. While Bayesian algorithm implemented in RELION is very powerful for high resolution 3D reconstruction, other 3D reconstruction algorithms and softwares (eg. Cryosparc, Freealign, EMAN, IMAGIC) exist and should be acknowledged.

3) The different terms in equation 2 should be explained.

4) The procedure for determination of vacuum intensity (I_0) was confusing and should be explained in greater detail.

5) The authors mention that the Apoferritin data was collected on ice having a thickness below 15 nm. It will be useful if the lower limit (if any) of the ice thickness used for data collection is mentioned. Additionally it will be informative to have the fractional intensity values corresponding to the upper and lower limits of the ice thickness range used for data collection.

6) Several groups have implemented the ALS approach for determining ice thickness during high throughput data collection (Rice et al. (2018) and Cheng et al. (2020) detail how ALS is implemented in Legimon and Rheinberger et al. (2019) details how ALS is implemented in EPU and SerialEM). As mentioned by the authors all previous implementations require experimental calibrations on each individual microscopes. However, a contrast between the different implementations that goes beyond the experimental calibrations will increase the be useful.

Reviewer #3 (Remarks to the Author):

Brown et al describe an easy-to-implement method for measuring ice-thickness during data collection for single particle cryo-EM. The manuscript has a nice flow beginning with introduction of the technical challenges in data collection pertaining to ice thickness, followed by description of existing methods for estimating the thickness and finally the implementation of their scheme for employing ice thickness

measurements. The method provides a more general solution to measuring ice-thickness and would not require installation of additional equipment like energy filter. The manuscript covers most of the important points, but can benefit from additional details as listed below.

1. The figure legends in general are not very descriptive. Although they are described in the text, a reader would benefit from having expanded figure legends associated with each figure. In particular-
 - a. Figure 2, although the aperture size is listed based on units of inverse length, it would be convenient to know what it refers to in microns as well. This would facilitate ease of use as that will be the metric most easily recognized by a general user during data collection operation.
 - b. Figure 3, labeling has to be clearer. In panel A, the overlap of the box inset describing what the color of particles refers to and plot of thickness is distracting and needs to be moved. In panel C, what do the individual measurements refer to? Are these with respect to individual particles?
2. The implementation of the script appears to be using the EPU package for data acquisition provided by Thermo Fisher. The authors should clarify if this is compatible only with EPU package. Although this is useful, notes on how this algorithm can be employed or adapted with other data acquisition platforms like Legikon or SerialEM needs to be discussed and described for broad applicability of this package.
3. Figure 2 could use with a linear regression plot using least-squares method for estimating accuracy of predicting ice-thickness across different voltages and different thickness. Although the plot is visually intuitive, a quantitative measure would make it even more compelling. It is clear that with increasing ice thickness the accuracy of estimation suffers.
4. Equation 2 describing the measured mean free path has to be expanded to describe what the individual terms of the equation denote. Authors should ensure all equations are adequately described in the text.
5. The technique described in this manuscript describes the method one would use with holey grids. How would these measurements change if one were to use a continuous support film like graphene, graphene oxide or amorphous carbon on top of the grid-foil?
6. Although the resolution improvement for apoferritin from 2.1 to 1.88 Angstroms is substantial. To accurately document improvements, the authors must include a comparison with prior reconstructions, at least with respect to particle number, B-factor, number of micrographs used for data processing and proportion of picked particles used for final reconstruction. The utility of using thin ice regions for obtaining higher resolution maps is clear, but such a comparison will help quantify the differences better. Ideally, the authors must make comparisons for the same protein, under similar data collection conditions without selecting for holes where ice is less than 15 nm and see if there is a difference. In addition, an equivalent subset of micrographs where the ice thickness is greater than 15 nm would also aid in quantifying resolution improvements. Subsets of particles adhering to the air-water interface may be greater with regions of thin ice. It would be interesting to see if there is a trade off with improved resolution due to thinner ice versus proportion of particles having to be discarded due to probable damage of particles at the air-water interface.

Dear Prof Manidipa Banerjee,

Thank you for considering our manuscript for publication, please see below for our responses to the reviewer comments and the details of how we have modified the manuscript. We are grateful to the reviewers for their suggestions which we feel have strengthened the work.

Kind regards,

Hamish Brown and Eric Hanssen

Reviewers' comments:

Reviewer #1 and co-reviewer (Remarks to the Author):

To: Manidipa Banerjee, PhD, Editorial Board Member, Communications Biology

RE: Measurelce: Accessible on-the-fly measurement of ice thickness in cryo-electron microscopy

Having appropriate ice thickness is important for achieving maximal resolution in cryo-EM. Finding and selecting the appropriate areas in the grid during TEM imaging is also important, as this determines the efficiency of the rather costly data collection sessions. In this work, Brown & Hanssen develop a new tool, Measurelce, that simulates ice thicknesses from first principles, and can be used in the screening and selection of regions of cryo-EM grids with suitable ice.

The major strength of this work is the authors' use of elementary scattering physics to simulate ice thicknesses in cryo-EM images. They benchmark their simulations using both ice channel and tomography methods, which are established methods for determining ice thicknesses, and find good agreement with their simulations. This is an elegant approach, and as the authors mention, bypasses the need to experimentally calibrate each microscope, which can be time consuming. Finally, a software tool, Measurelce, is developed for visualizing ice thicknesses. Overall, the paper is well considered, meticulous, and accessible, and should provide another tool for cryo-EM scientists to estimate ice thicknesses. I recommend accepting with minor revisions.

I have the following comments and questions:

1. Since energy filters are widely in use on many Krioses, can the authors comment on the applicability of their simulations to such systems? In Rice et al 2018 the ALS coefficients for a Krios with and without an energy filter and at different slit widths are quite different, so one might expect this to influence the simulated ice thicknesses?

The overall goal of our study is to mainly inform the user on ice thickness on screening microscopes and inform the acquisition software inbuilt histogram filtering of target areas. These measures can be done without the energy filter. However, we thank the reviewer for the suggestion and would envisage to implement this in subsequent version of Measurelce.

Since Measurelce calculates the fraction of inelastically scattered electrons at any given thickness using experimentally measured values of the inelastic mean-free path, the Measurelce simulations could be tweaked to remove inelastically-scattered electrons from the simulation in manner analogous to the energy filter slit. The onset of the low-loss peak for ice is around 7 eV¹ so these calculations would then be valid for a well-centred slit with a width of around 10 eV which would remove almost all inelastically scattered electrons from the image. We suspect that achromaticity, ensuring there is no noticeable energy dispersion across the field of view, would be challenging for most systems at the magnifications that are best for ice thickness screening (2,000 – 10,000 times magnification) at this narrow a slit width. For confidence in this approach a similar benchmarking process as that in Fig.2 of the paper would then be necessary so we suggest this as a possibility for future release of Measurelce. Extending the Measurelce approach so that any energy filter slit width can be used would be a larger undertaking. The current approach assumes a single mean-free path to estimate the proportion of electrons that lose *any* amount of energy for a given thickness of ice. For a wider energy-filter slit that would exclude only a subset of the energy loss spectrum from the TEM image, a simulation might need to consider different mean free paths at different energy losses – though the existing approach might still be sufficient. This case is considerably more complex than the <10 eV slit approach discussed in the previous paragraph so is a problem for future research.

2. In Figure 3 the Measurelce vs tomography ice thickness starts to deviate from ideal at > 80 nm. Why is this?

We can think of two possible explanations for this:

1. There is some defocus induced contrast at the edge of the foil holes
2. Since pure water samples are assumed in the simulated lookup tables, we note in the manuscript that Measurelce might underestimate ice thickness for real protein samples. Whilst the simulations are valid for samples less than 80 nm – the desired ice thickness for SPA samples - we start to see some deviation from expected results for thicker samples.

Explanation #1 wouldn't explain the underestimate for the uniformly thick hole – tomogram 3 – so we think #2 is the more probable cause of this effect. We have updated the text to point out the discrepancy and suggest our hypothesis for its cause.

3. Is Measurelce integrated with the EPU workflow? Or is it check out actual ice thickness values with Measurelce ∅ Compare holes in Measurelce to EPU ∅ set EPU ice filters. The way Measurelce is used for "on-the-fly" screening was not immediately clear to me.

Measurelce isn't integrated into EPU since EPU is proprietary, closed-source software so the latter approach (a manual method of using Measurelce to identify an appropriate intensity that is then used as an "ice-filter" cutoff in EPU) was used for selecting holes for the equine apoferritin collection. The term "on-the-fly" in the manuscript refers to how the software can be installed on the microscope control or support PC to analyse micrographs during experiment. With modifications

¹ Shibaguchi, Takashi, Hideo Onuki, and Ryumyo Onaka. "Electronic structures of water and ice." *Journal of the Physical Society of Japan* 42.1 (1977): 152-158.

made in response to comments from reviewers #2 and #3, closer integration into the single particle analysis workflow is now available for Legion and SerialEM (using a plugin developed by other Rheinberger et al.²) but integration within EPU would only be possible with input from Thermo-Fisher scientific and might prevent users of other electron microscope brand from using Measurelce so we persist with open source microscope automation software for the time being.

4. The authors recommend hole-magnification images to be taken at close to focus to reduce defocus-induced phase contrast. In practice, these images are routinely taken at ~-50 μm defocus for better contrast for hole finding. How do the authors balance the need for defocus-induced contrast for hole finding with the need to be close to focus for accurate ice thickness measurements?

Though contrast is of course reduced in comparison to a ~-50 μm defocus image, an objective aperture on its own produces sufficient contrast for identifying holes both visually and automatically with EPU, see eg. Fig. 3 (d). This is helped by the fact that at the low magnifications used for grid screening a large dose per camera pixel can be used that will translate to a negligible (<0.01 e/Å²) dose/Angstrom squared.

5. Under the section "Uncertainty in literature values of inelastic mean free path", there is this sentence: "However it must be acknowledged that the inputted value of inelastic mean-free path is the most obvious source of error in the Measurelce calculations." It wasn't quite clear to me what this meant, as inelastic mean-free path does not appear to be a direct input option in Measurelce.

Inelastic mean-free paths are stored internally within the software so this parameter isn't required to be inputted by the user. We've adjusted the text to say this explicitly. There is the option of manually inputting a user provided value for the inelastic mean free path, that was used for example to explore how uncertainty in inelastic mean-free path would effect ice measurement in Fig. 4(b), but end users of Measurelce are not encouraged to make use of this feature.

6. It took me several tries to digest the section "Uncertainty in literature values of inelastic mean free path". If the authors would write a sentence or two at the start of that section to summarize the problem, I think that would help a lot with setting the context.

We agree with the reviewer that better context needs to be given for this section so have rearranged it to summarise the problem and conclusion at the beginning of the section before the text discusses the technical details of the problem.

7. In my opinion the authors should move the final sentence in "Using Measurelce for screening and data collection" into "Conclusions". The sentence reads: "This result does however make the case for including quantitative ice thickness tools like Measurelce within cryo-EM screening and acquisition workflows, especially considering the convenience of the Measurelce software tool." A 1.88 Å

² Rheinberger, Jan, et al. "Optimized cryo-EM data-acquisition workflow by sample-thickness determination." *Acta Crystallographica Section D: Structural Biology* 77.5 (2021).

reconstruction of apoferritin on its own, without the appropriate controls, does not make a case for ice thickness curation. However, the case is made when considering the story in its entirety and in context of the existing literature, which the authors have thoroughly explained in the full text.

We think this a reasonable suggestion and have done as the reviewer has asked..

And a few additional comments:

- It is odd to describe SPA as a characterization tool.

We have changed the word “tool” for “technique”

Also perhaps refer readers to a nice recent review in the first lines of the Introduction section.

We have added a citation, in the introduction, to:

Lyumkis, Dmitry. "Challenges and opportunities in cryo-EM single-particle analysis." *Journal of Biological Chemistry* 294.13 (2019): 5181-5197.

That might be more helpful than other somewhat random details provide (e.g. the dose tolerance from Grant and Grigorieff which I think is no longer the accepted norm).

Grant and Grigorieff, only consider larger (>300 kDa) proteins and find a dose of 20 e/A² to be superior to a dose of 53 e/A² though they note that higher doses might be beneficial for smaller proteins. Danev et al.³ have since rigorously confirmed that this is indeed the case for a high dose (65 e/A²) acquisition of a small (~120 kDa) protein. Since the statement in our manuscript was mainly about extra dose reducing resolution by inducing damage we have chosen to just leave this statement with the single citation to Grant and Grigorieff and consider the referees criticism as mainly addressed by the addition of the citation to the Lyumkis review.

Similarly to quote a paper from 2012 describing resolution capabilities seems inappropriate.

We have moved the citation to the middle of the sentence so that it reads: “...sophisticated algorithms (Scheres, 2012) combine tens or even hundreds of thousands...” to make it clear that this citation is purely a reference for these algorithms, not a statement about current and future achievable resolutions.

As is the term “recent” for describing direct detectors.

This is a reasonable proposition, especially given that the current speed of advancement in cryo-TEM means that advances made prior to even 2017 seem old so we have deleted the “recent” descriptor.

³ Danev, Radostin, et al. "Routine sub-2.5 Å cryo-EM structure determination of GPCRs." *Nature Communications* 12.1 (2021): 1-10.

- It would have been interesting to see a comparison of the results of the MeasureIce method to those obtained from an energy filter using the Method described by Rice et al.

Similar to the situation with the ALS method prior to this paper, the energy filter method has to be experimentally calibrated so measurements using this method would not be independent of either tomography or the “ice-channel” method.

- Could the authors comment on the disparity between the ice thickness ranges that are measure for the tomography tests (between 20 and 100nm) vs. the hole drilling test (much thicker and few below 100nm).

As noted earlier we think this is most likely due to a real protein sample being used in the former benchmark and a purified water being used in the latter. We’ve changed the wording at the end of the tomography to reflect this.

- I do not think that 2.1 Å is any longer the best available resolution for ApoF.

It depends on the specific apo-ferritin, as detailed in Nakane et al. Mouse apo-ferritin has been resolved down to 1.22 Angstrom but 2.1 Å was the best resolution we could find in the literature for Equine apo-ferritin provided by Sigma. For some reason there seem to be some very large difference between the difference source of ApoF. Hence we only claim an improved resolution for this particular source of protein.

Reviewer #2 (Remarks to the Author):

Brief summary of the manuscript:

Over the last 10 years single particle cryo-electron microscopy (cryo-EM) has become a mainstay technique in structural biology and this method has been used to solve several high resolution structures. Despite the fantastic progress made to streamline cryo-EM data collection and subsequent image processing and 3-D reconstructions, several bottlenecks remain. One of the major bottle is identification of which areas of a vitrified grid will be suitable for data collection for high resolution structure determination and this step often requires the input of expert electron microscopists. Whether micrographs have been collected from suitable areas of the grid often becomes apparent after the data collection session and this may lead to wastage of valuable microscope and user time. One of the major consideration for identifying “good” areas on a grid is the optimal thickness of the vitreous ice. If the ice is too thin then particles will not be embedded in the ice and too thick an ice layer will contribute to added background noise. In this work the authors have formulated a standalone ice thickness determination software called MeasureIce. This software can be used on all microscopes and does not require any experimental calibration. Using Apoferritin as a test sample, the authors demonstrate MeasureIce guided selection of grid holes can lead to high resolution reconstruction.

Overall impression of the work:

This work has the potential to streamline cryo-EM data collection by introducing the objective metric of ice thickness as a guide for selecting good areas of a grid. Moreover by reducing the dependence on an expert electron microscopist for data collection, Measurelce might aid in further democratization of cryo-EM. The ease of adapting Measurelce to different electron microscopes hints at the wide use and versatility of this tool. However, a few experiments and clarifications (detailed below) would increase the impact of the work.

Major comments:

1) According to Rheinberger et al. the apparent mean free path (Teff) depends on a number of factors including the magnification regime of the microscope (for instance LM, M or SA mode for a FEI/TFS microscope). If that is indeed the case then it must be mentioned what magnification regime is optimal for using Measurelce and if Teff is not dependent on the magnification then an explanation should be provided about the discrepancy between the author's assumption and the previous literature.

Since most of the work for this manuscript was performed on two-condensor systems, which dictate using the "selected-area" (SA) magnification mode for a parallel illumination condition, we had not considered the fact that, of course, a three-condensor lens TEM allows a parallel beam in LM mode. The magnification mode changes usually involve turning whole sets of lenses either on or off (eg. The objective lens is turned off in low mag). This will affect the magnification of the electron diffraction pattern formed in the back focal plane immediately (a few millimetres) below the sample, where the aperture strip is inserted. This change in back-focal plane "magnification" will affect the size of the objective lens as measured in diffraction plane units of inverse Angstrom or mrad meaning that a greater or fewer number of the scattered electrons will be removed from the TEM image by the objective lens. Thus the image might appear darker or lighter for a given thickness of ice, and this is why the slope of I/I_0 vs thickness changes, that is to say T_{eff} has changed.

2) A detailed description of how Measurelce was used during a data collection session would be useful for microscopists planning to use the software. The following information will be useful for potential users:

- a. Whether there is any particular microscope alignment (other than the direct alignments performed routinely) that is crucial for the success of Measurelce*
- b. At which stage ice thickness was determined (while selecting squares from grid atlas or selecting holes in a grid square or selecting exposure area in a hole)*
- c. Was Measurelce used once during the data collection session to figure out the intensity values corresponding different ice thicknesses or was ice thickness measured at multiple points (for instance while selecting holes within each grid square)*
- d. What magnification, beam condition etc. were used for acquiring the image that acted as the input for Measurelce*

We have added a section titled “Recording images for Measurelce on the TEM” to the manuscript.

3) As has mentioned by the authors, the high resolution reconstruction of the Apoferritin test sample might not be solely due to the use of Measurelce. It would be useful to clearly demonstrate whether ice thickness can indeed be used as an objective metric for collecting high quality micrographs. Using Measurelce the authors might collect images with optimal ice condition (<15 nm) and with different suboptimal thicknesses of ice (for instance 30 nm, 50 nm etc). 3D Reconstructions calculated with particles from different ice thicknesses might then be compared (in terms of resolution, B-factor etc.) to determine whether ice thickness can be used as the sole metric for selecting suitable areas of a grid.

We have added just such a table to the manuscript, a separate issue that makes “apples-to-apples” comparison difficult is that the apoferritin reconstruction from Naydenova et al. was as a demonstration of functionalized graphene coated grids. We note this in the text of the manuscript.

Since we have submitted this abstract another cryo-EM group from the Diamond Light Source Electron Bio-imaging centre have answered the second question posed by the reviewer quite elegantly⁴. By partitioning each of their TEM images according to ice thickness (as estimated by background TEM image intensity) they solve 3D structures from particles extracted from regions of similar ice thickness. Thinner ice leads to a clear and substantial improvement in a T20s dataset (EMPIAR-10025) despite thicker ice having more top-views of that particle. For a gamma secretase dataset (EMPIAR-10194) thicker ice yielded the highest resolution maps with the authors attributing the worsening resolution to a more uniform distribution of orientations being present in the thicker ice. The conclusion then is that though ice thickness leads to better signal to noise, so an “objective” measure in that sense, this isn’t the sole metric of importance in cryo-EM screening and acquisition – the user still needs to check that the particles do not exhibit preferred orientations. To acknowledge this fact we have added the sentence:

“We also note that in other work thinner ice has been shown to lead to higher resolution 3D maps for symmetric particles, but thinner ice can lead to a worsening of preferred orientation that can degrade the reconstruction resolution for asymmetric particles”

with a citation to Olek et al in the section detailing the apoferritin reconstruction in our paper.

Minor comments:

1) It will be useful to comment on whether Measurelce can be used for grids having a support layer of carbon or graphene over the holes.

⁴ Olek, Mateusz, et al. "IceBreaker: Software for high-resolution single-particle cryo-EM with non-uniform ice." *Structure* (2022).

We suspect a thin layer of graphene will have minimal effects on ice thickness measurements since a few monolayers of carbon would be expected to have minimal effects on beam intensity. Thicker carbon films will have a greater effect and this comment by the referee has sparked some investigation on our part that have convinced us of a very straight forward solution to the measurement of ice thickness with a backing carbon film.

We simulated ice thickness image intensity calibration curves using measureice and modified the code to incorporate a carbon films of 10 nm thickness atop the ice. We used a realistic atomic model of amorphous carbon⁵ and theoretically predicted values of inelastic mean free path⁶. Simulating a 200 keV TEM with 10 mrad and 15 mrad apertures and plotting just ice thickness against image intensity, the latter on a log scale, shows us that the addition of a carbon film simply shifts the intercept of the resulting lines and the effect on the slope, which is the ALS coefficient, is minimal. In other words it is trivial to adapt measureice calibration curves to carbon backed samples by choosing the reference intensity (I_0) to be the intensity measured for the carbon backing (instead of vacuum).

2) In page2 the authors mention that Bayesian algorithms are used for image processing. This is not

⁵ Ricolleau, C., et al. "Random vs realistic amorphous carbon models for high resolution microscopy and electron diffraction." *Journal of Applied Physics* 114.21 (2013): 213504.

⁶ Shinotsuka, H., et al. "Calculations of electron inelastic mean free paths. X. Data for 41 elemental solids over the 50 eV to 200 keV range with the relativistic full Penn algorithm." *Surface and Interface Analysis* 47.9 (2015): 871-888.

completely accurate. While Bayesian algorithm implemented in RELION is very powerful for high resolution 3D reconstruction, other 3D reconstruction algorithms and softwares (eg. Cryosparc, Frealign, EMAN, IMAGIC) exist and should be acknowledged.

The reviewer is correct and we have changed the text and removed that sentence in order to be more inclusive of all software used.

3) The different terms in equation 2 should be explained.

We have added the following sentence: “Where M_w , ρ and Z , are the molecular weight, density and atomic number of the sample and U_0 , m , c and β are the beam energy, electron mass, speed of light and electron speed (as a fraction of c)”

4) The procedure for determination of vacuum intensity (I_0) was confusing and should be explained in greater detail.

We have expounded this in greater detail in the revised manuscript.

5) The authors mention that the Apoferritin data was collected on ice having a thickness below 15 nm. It will be useful if the lower limit (if any) of the ice thickness used for data collection is mentioned. Additionally it will be informative to have the fractional intensity values corresponding to the upper and lower limits of the ice thickness range used for data collection.

There was no lower limit for the ice thickness and some holes evidently had a thickness less than the diameter of apoferritin since no particles are visible in those micrographs. We also acknowledge that some holes were likely above the 15 nm criterion due to Poisson noise in our low magnification images increasing the apparent ice thickness.

6) Several groups have implemented the ALS approach for determining ice thickness during high throughput data collection (Rice et al. (2018) and Cheng et al. (2020) detail how ALS is implemented in Leginon and Rheinberger et al. (2019) details how ALS is implemented in EPU and SerialEM). As mentioned by the authors all previous implementations require experimental calibrations on each individual microscopes. However, a contrast between the different implementations that goes beyond the experimental calibrations will increase the be useful.

Our approach is *identical* to that in Leginon and the plugins for SerialEM and Digital Micrograph developed by Rheinberger et al. with the exception that we use simulations to perform the tedious calibration component. Both approaches require that users input the effective inelastic mean free path for their setup into Leginon or the SerialEM plugin and we have modified the `Generate_Measureice_calibration.py` script to give these values to the user, noting this in the manuscript. We like the simplicity of the MeasureIce program in that it only does ice thickness

measurement so think an approach were we can integrate with more detailed software packages that allow live data collection workflows on the microscope whilst keeping Measurelce as a separate entity is a “best of both worlds” solution.

Reviewer #3 (Remarks to the Author):

Brown et al describe an easy-to-implement method for measuring ice-thickness during data collection for single particle cryo-EM. The manuscript has a nice flow beginning with introduction of the technical challenges in data collection pertaining to ice thickness, followed by description of existing methods for estimating the thickness and finally the implementation of their scheme for employing ice thickness measurements. The method provides a more general solution to measuring ice-thickness and would not require installation of additional equipment like energy filter. The manuscript covers most of the important points, but can benefit from additional details as listed below.

1. The figure legends in general are not very descriptive. Although they are described in the text, a reader would benefit from having expanded figure legends associated with each figure. In particular- a. Figure 2, although the aperture size is listed based on units of inverse length, it would be convenient to know what it refers to in microns as well. This would facilitate ease of use as that will be the metric most easily recognized by a general user during data collection operation.

We have included the values in microns as well though impress upon the reader that this number *only* gives the size of the physical hole in the aperture strip, the TEM objective lens focal length determines the size of this cutoff with respect to the diffraction pattern formed in the back focal plane and this is the quantity of interest. The generate_Measurelce_calibration.py script already allows users to label calibration curves using microns for ease of application.

b. Figure 3, labeling has to be clearer. In panel A, the overlap of the box inset describing what the color of particles refers to and plot of thickness is distracting and needs to be moved. In panel C, what do the individual measurements refer to? Are these with respect to individual particles?

We have adjusted the Figure so that the legend in Fig 3(a) is below the plot. Individual data points in Fig. 3(c) correspond to single pixels in the Measurelce thickness map, eg. The one shown in Fig 3(d), and we have now described this in the figure caption.

2. The implementation of the script appears to be using the EPU package for data acquisition provided by Thermo Fisher. The authors should clarify if this is compatible only with EPU package. Although this is useful, notes on how this algorithm can be employed or adapted with other data acquisition platforms like Leginon or SerialEM needs to be discussed and described for broad applicability of this package.

A similar comment was brought up by reviewer #2. Measurelce is currently a standalone package and its results are used to inform the acquisition software. We have modified Measurelce to output

the effective mean free path, often referred to as the ALS coefficient, which can then be easily inputted into Legimon and SerialEM (the latter using the plugin built by Rheinberger et al.).

3. Figure 2 could use with a linear regression plot using least-squares method for estimating accuracy of predicting ice-thickness across different voltages and different thickness. Although the plot is visually intuitive, a quantitative measure would make it even more compelling. It is clear that with increasing ice thickness the accuracy of estimation suffers.

We have added a Pearson correlation (R^2) coefficients to these plots, with explainer text.

4. Equation 2 describing the measured mean free path has to be expanded to describe what the individual terms of the equation denote. Authors should ensure all equations are adequately described in the text.

We have rectified this – reviewer #2 pointed out the same omission, see minor comment #3.

5. The technique described in this manuscript describes the method one would use with holey grids. How would these measurements change if one were to use a continuous support film like graphene, graphene oxide or amorphous carbon on top of the grid-foil?

Reviewer #2 has also asked this question see minor comment #1 for our detailed response.

6. Although the resolution improvement for apoferritin from 2.1 to 1.88 Angstroms is substantial. To accurately document improvements, the authors must include a comparison with prior reconstructions, at least with respect to particle number, B-factor, number of micrographs used for data processing and proportion of picked particles used for final reconstruction. The utility of using thin ice regions for obtaining higher resolution maps is clear, but such a comparison will help quantify the differences better. Ideally, the authors must make comparisons for the same protein, under similar data collection conditions without selecting for holes where ice is less than 15 nm and see if there is a difference. In addition, an equivalent subset of micrographs where the ice thickness is greater than 15 nm would also aid in quantifying resolution improvements. Subsets of particles adhering to the air-water interface may be greater with regions of thin ice. It would be interesting to see if there is a trade off with improved resolution due to thinner ice versus proportion of particles having to be discarded due to probable damage of particles at the air-water interface.

A similar general point is raised by reviewer #2 in remark 3 so we address this in detail there. The reviewer's comment that the effects of an air-water interface are worse in regions of thinner ice is an intriguing one that warrants some discussion. Olek et al. found that thin ice can make things worse via a worsening of preferred orientation so that would appear to be the most common scenario in which thin ice might not be desirable. In our opinion the best study of air-water interface

related questions is by Noble and colleagues⁷, with the usual caveat that results might be different for different proteins, they found that where particles adhered to an air-water interface (as most proteins in their study did) they did so regardless of ice thickness. Noble et al. didn't find an association with thin ice and air-water interface damage though this was not a question they asked so their study cannot rule this out.

Note that using cryosparc's "relative ice thickness" metric we found no association between ice thickness and excluded particles (those belonging to poor quality 2D-classes), suggesting that this effect is not present in our sample.

⁷ Noble, Alex J., et al. "Routine single particle CryoEM sample and grid characterization by tomography." *Elife* 7 (2018): e34257.

REVIEWERS' COMMENTS:

Reviewer #1 (Remarks to the Author):

We would like thank the authors for the detailed rebuttal. We feel that our concerns have been sufficiently addressed by the authors.

Reviewer #2 (Remarks to the Author):

The authors have addressed all my comments and suggestions. I have no further comments.

Reviewer #3 (Remarks to the Author):

Minor comments-

1. In "Amorphous Carbon and Graphene Backed section" replace IO with Io.
2. Figure 3 legend replace 'Tthe' with 'The'
3. In the methods section 1.52 second total exposure with 7 frames implies uneven frame fractionation rate. Please clarify if this is so and if so what is the dose distribution across frames and how was this accounted for during data processing.
4. Was symmetry expansion performed during data processing of apoferritin?

1. In "Amorphous Carbon and Graphene Backed section" replace IO with Io.

We thank the reviewer for noticing this, we found another instance of this typo in the caption of Figure 5 and fixed that as well

2. Figure 3 legend replace 'Tthe' with 'The'

We have done as requested

3. In the methods section 1.52 second total exposure with 7 frames implies uneven frame fractionation rate. Please clarify if this is so and if so what is the dose distribution across frames and how was this accounted for during data processing.

We thank the reviewer for the comment about unevenness of the electron dose, we have changed the text to reflect the process which we didn't make clear in our first attempt.:

“Exposures of 1.52 s were taken with a fluency of 5.34 e/px/s resulting in a total dose of $20e^{-}/\text{\AA}^2$ on the sample with 7 frames per movie”

to

“Exposures of 1.52 s were taken with a fluency of 5.34 e/px/s resulting in a nominal dose of $\sim 20 e^{-}/\text{\AA}^2$ on the sample with 8 frames per movie. The last frame was deleted to avoid unevenness of the dose distribution resulting in 7 frame movies with a total dose of $\sim 16 e^{-}/\text{\AA}^2$ ”

4. Was symmetry expansion performed during data processing of apoferritin?

Octahedral symmetry was enforced in 3D refinement of apoferritin (we now clearly state this) but at no point were the particles themselves symmetry expanded.